# Effects of Soundscape on Flow State during Diabolo Exercise

**DOI:** 10.3390/ijerph19138034

**Published:** 2022-06-30

**Authors:** Tong-Yu Li, Si-Yuan Guo, Bin-Xia Xue, Qi Meng, Bo Jiang, Xin-Xin Xu, Chein-Chi Chang

**Affiliations:** 1School of Architecture, Harbin Institute of Technology, Harbin 150006, China; li_tonghe@126.com (T.-Y.L.); mengq@hit.edu.cn (Q.M.); jiangbo02246006@163.com (B.J.); chang87@gmail.com (C.-C.C.); 2Key Laboratory of Cold Region Urban and Rural Human Settlement Environment Science and Technology, Ministry of Industry and Information Technology, Harbin Institute of Technology, Harbin 150001, China; 3School of Civil and Architecture Engineering, Northeast Petroleum University, Daqing 163319, China; xinxin_xu@outlook.com

**Keywords:** diabolo, exercise psychology, flow state, soundscape, mental health, older adults

## Abstract

“Soundscape” is defined as “an acoustic environment as perceived or experienced and/or understood by a person or people, in context.” The research focuses on the physical properties of sound, paying attention to the relationship between the soundscape and people. Sports provide a comprehensive context, including the athletes, their behavior, the environment, spectators, and other elements. Soundscape in sports has many important functions, such as indicating the movement process, promoting positive emotions, and enhancing the entertainment atmosphere. It is also related to the physical and mental health of people who are exercising. As a technical sport with multidimensional soundscape qualities, diabolo is very popular among older adults in China due to its easy portability and operation. In order to explore the influence mechanism of the soundscape on the mental health of older players and better guide the shaping of soundscapes at sports venues, this paper focuses on soundscape perception and proposes a two-channel (direct and indirect) model of the influence of the diabolo soundscape on the psychological perception of older players. At the same time, we introduce the flow state as an indicator to evaluate mental health, and use the Short Flow State Scale (SFSS) to measure the flow state. By setting up a control experiment using the diabolo with and without sound and using the two-channel (direct and indirect) influence model to compare the differences in flow state scores under the two states, we analyzed the influence mechanism of the diabolo soundscape on the flow state of elderly groups. The results show that the scores of various flow state dimensions and total flow scores in the silent state of diabolo exercise are significantly lower than those in the sound state (*p* < 0.05 and *p* < 0.001), and the main differences are in the three dimensions of unambiguous feedback, sense of control, and autotelic experience. The two-channel influence model can effectively explain the differences in older adults’ flow state, which provides a new comprehensive perspective to study the effect of the soundscape in sports on people’s mental health.

## 1. Introduction

The soundscape represents an emerging discipline examining the interconnections between sound, users, space, and the environment [1]. In 2014, the International Organization for Standardization (ISO) defined “soundscape” as “an acoustic environment as perceived or experienced and/or understood by a person or people, in context” [2]. Soundscape refers to both natural and human-created environmental sound; it also includes listeners’ perception of sound in the environment [3]. Soundscape research needs to focus on “how the sound environment is understood by a person living in the environment” and how to regulate the relationship between people, the environment, and sound accordingly [4]. Most studies to date have focused on the effects of noise and its negative impact in cities, and less on desired and preferred sounds and the positive emotions they generate. With the development of disciplinary integration, soundscape research has begun to incorporate human perception as a factor, centering on the positive outcomes of soundscapes (especially in terms of mental health) [5]. Scholars have conducted many studies in this area, some of which have focused on the relationship between mental health and the soundscape [6]. The results showed that the experience of a pleasant soundscape facilitates faster recovery from stress [7]. Birdsong in urban parks can improve visitors’ perceived levels of mental restoration [8]. A quiet rural soundscape could be of benefit to the general mental health of the population due to its potential for psychological restoration [9]. Nature, light music, and symphonic soundscape elements promote people’s physical and mental health, while traffic soundscape elements have the opposite effect [10].

At present, research on the soundscape is mostly concentrated in the fields of architecture, environmental science, psychology, etc. [11,12]. Even though there is less research on the soundscape in sports, it plays an important role. Kerrigan et al. noted that runners use mobile technology to create musical soundscapes in order to enhance their running experience, and these soundscapes play an essential role in providing a pleasant experience while running [13]. Within the sports science literature, it is acknowledged that a soundscape with a musical focus can mediate between the body and mind, and therefore has not only a physical but also a psychological impact on an athlete’s performance [14]. At the same time, the soundscape is a critical indicator for athletes. For example, visually impaired cricket players can use sound to measure the speed, rotation, and direction of the ball [15]. Noise problems at sports venues have also been investigated. Lee et al. measured the noise levels at a baseball stadium and analyzed fans’ attitudes on the effect of recreational noise exposure on their hearing. Most respondents reported that they did not consider wearing earplugs, and one-third experienced hearing problems after the game [16]. Cranston et al. evaluated occupational and recreational noise exposure at two sporting arenas hosting hockey games, and the relevant data reflected the potential for overexposure at indoor hockey events [17]. 

In summary, there is little research on soundscapes in sports at present; the research mainly focuses on how to improve the soundscape, promote the performance of athletes, and reduce the impact of bad sounds (mainly noise on the field). There is a lack of research on the psychological impact of the soundscape on exercisers. The soundscape includes the complete sound environment at a site and the human response to it [18]. Therefore, the study of soundscapes should focus not only on the specific place (such as park or street), but also on the relationship between the soundscape and people. Sports is a comprehensive context including athletes, their behavior, the sports environment, spectators, and other elements. In this context, exploring the mechanism of the impact of the soundscape on people’s mental health is the gap that this paper seeks to fill.

Diabolo is a good game with a multidimensional soundscape. It is popular among the elderly due to its moderate intensity, recreational nature, and capacity to promote physical and mental health. The classical diabolo is made of wood. As shown in the picture in Figure 1, the diabolo has two main parts: the shaft and the wheel. The wheel is surrounded by whistles of different sizes, with a large whistle for the bass and a small whistle for the treble. The person winds the rope around the shaft and makes the diabolo spin by swinging their arm up and down. When the diabolo is spinning, air enters the wheel through the whistles and flows through a small, semi-circular air chamber surrounded by thin wooden reeds; the air then flows out, emitting a loud sound. Plastic diabolos are also common today; while they are improved in terms of resistance to breakage and portability, they do not have whistles, so they do not make sounds. The three-view size chart of the diabolo in Figure 2 shows that its shape and size are moderate, and using it provides a visual and tactile experience.

To further investigate the impact of the soundscape on people’s mental health while exercising, the concept of the flow state is introduced. A flow state is an optimal psychological state [19] in which one is fully engaged in an activity and experiences many positive feelings, including the freedom of self-awareness and great enjoyment of the process itself [20]. Sport is a major source of the flow state, with competitive sport and physical exercise being the most likely domains in which one can attain it [21]. The flow state is significantly positive associated with levels of psychological well-being and is frequently associated with increased well-being [22], positive subjective experience [18], and objective performance [23]. Therefore, improving the flow state has a positive effect on the mental health of athletes/exercisers.

The origin of the flow state can be traced back to the humanistic psychologist Maslow’s “peak experience”, which is an overwhelming sense of enjoyment and pleasure resulting from an individual’s total engagement in an activity. The phenomenon of peak experience in sports was noted in Ravizza’s research [24]. Csikszentmihalyi specified the flow state as an important positive emotion, a special state of self-enjoyment that occurs naturally, during which the individual is fully engaged in the activity in which he/she is involved [25]. Later, he proposed nine psychological characteristics of flow state: challenge–skill balance, action–awareness merging, clear goals, unambiguous feedback, concentration on the task at hand, a sense of control, the loss of self-consciousness, the transformation of time, and the autotelic experience [26]. His book *Flow: The Psychology of Optimal Experience* was the first work to provide a systematic discussion of the flow state [27]. 

Jackson and Marsh introduced the flow state to the field of sports psychology in 1996. They defined it as an optimal experiential state in which an athlete is fully engaged in a task, creating a state of consciousness that allows performing at an optimal level of athleticism [20]. They developed the Flow State Scale (FSS), which contains 36 questions in 9 dimensions and is one of the most widely used flow state measurement scales. Based on this, Jackson and Eklund developed the revised and improved FSS-2 [28], but the excessive number of entries increased the burden on respondents, which affected the reliability of collected data. In 2008, Jackson, Martin, and Eklund redeveloped the FSS, this time creating the Short Flow State Scale (SFSS) to provide a more convenient measurement tool for the flow state in sport [29]. The cross-cultural validation and revision for the Chinese version by Liu Weina confirmed the universality of flow state theory and showed that the scale has good reliability and validity [30].

To the best of the authors’ knowledge, no review has been performed so far on the association between the soundscapes of sports and the mental health of exercisers. Thus, the aim of this paper was to explore such a relationship. For this purpose, we chose the flow state as an important indicator of mental health and measured it with related scales, and selected diabolo (a technical sport in China with multidimensional soundscape qualities) as the research object. Through field research and soundscape perception experiments, we intended to fulfill the following aims: (1) to construct a two-channel (direct and indirect) influence model of soundscape perception when playing diabolo, and (2) to compare the differences in flow state of the elderly playing diabolo with and without sound using the constructed model to explain the reasons for the differences, in order to provide a new comprehensive perspective to study the effect of the soundscape in sports on people’s mental health and make suggestions for the shaping of soundscapes at sports venues.

## 2. Materials and Methods

### 2.1. Direct and Indirect Influence Model of Diabolo Soundscape Perception

The physical parameters of a sound scene (frequency, duration, timbre and intensity, etc.) often have a direct impact on people’s psychological perception, stimulating the auditory system. At present, there are many studies on the impact of the soundscape’s physical parameters on psychological perception. For example, a study on the relationship between loudness and pleasantness showed that the pleasantness of stimuli at intermediate volume levels is not influenced by the volume, but for sound at relatively high volume, there is a good correlation between the two [31].

The human reaction to sound is not just a physical perception, but also an aesthetic sensation in response to the environment [32,33]. The soundscape has deep symbolic meaning, as when people associate the sound of birds with a beautiful natural environment or even a beautiful picnic experience. This indicates the deep indirect impact of the soundscape on people’s psychological perception. Such perception is often not direct, but the result of experience, emotion, and the soundscape itself.

Based on the above two influencing factors and the soundscape characteristics of diabolo, we developed a two-channel (direct and indirect) influence model of diabolo soundscape perception (Figure 3). The first is the direct effect of the physical characteristics of the diabolo soundscape (frequency, duration, timbre, and intensity) on psychological perception. Figure 4 shows the five basic steps of diabolo exercise, namely A-Start; B-Accelerate; C-Swing; D-Throwing and Catching; E-Finish, and Figure 5 shows the change of diabolo sound pressure level with time in the five steps. The sound of diabolo is loud, close to the buzzing sound of metal, rhythmic, and moderate in intensity (according to the change trend over time shown in Figure 5, the range of sound pressure levels of diabolo is about 55–90 dB), which can stimulate the auditory nerve and create a good experience while using it, just as birdsong and light music can bring enjoyment.

The second channel is the indirect perception of the diabolo soundscape. First, the sound plays a role in coordinating movement. Using the diabolo involves a series of movements as a whole, and sound is an important component connecting the links. Second, the sound plays an indicator role. As can be seen from Figure 4 and Figure 5, the diabolo soundscape differs in different stages of movement (starting, accelerating, swinging, etc.). By perceiving the soundscape, users can clearly know the steps of the exercise, which allows them to prepare for the next step. Third, the diabolo soundscape is a sign of success for users. The diabolo has a certain technology, so that its “voice” or sound indicates the user’s success, which is a realization of self-value. Fourth, the sound will attract pedestrians to stop and watch, which brings a kind of psychological reinforcement and enhances the expression and social desire of exercisers, so that they will have enhanced self-efficacy and self-esteem through self-presentation.

### 2.2. Evaluating the Flow State

To evaluate the flow state, more authoritative tools include the FSS and the SFSS. We integrated the advantages of both and applied Liu Weina’s Chinese version of the SFSS. The scale contains 9 dimensions, with one question for each dimension (Table 1). The SFSS has high reliability, and many scholars use it to study people’s flow state while engaging in physical exercise [34]. It employs a 5-point Likert scale (1 = does not conform at all to 5 = conforms very much) to assess the flow state of elderly people doing diabolo exercises. The higher the score, the higher the flow state attained. We evaluated the flow state during diabolo exercises to measure the impact on promoting mental health.

### 2.3. Control Experiment of Diabolo Soundscape Perception

#### 2.3.1. Experimental Site

In 2011, the diabolo-shaking game was listed as an intangible provincial cultural heritage in Jiangsu Province, China; the declared area of origin was Nanjing, the capital of the province, which has a deep cultural heritage of this activity and a wide range of groups. Based on preliminary interviews and field surveys, we used the green park space in front of Nanjing Xuanwu Gate as the research site to gauge the sound source perception of diabolo shaking and to investigate the flow state among users. The park is overall a regular rectangle, with a length of approximately 283 m, a width of about 89 m, and a green coverage rate of 93%. There are groups of older adults who regularly participate in the diabolo-shaking sport. Most groups are intentionally scattered to maximize the activity space, reduce the interference of diabolo sounds, and lower the risk of injury due to improper handling.

#### 2.3.2. Experimental Subjects

In China, the main group playing diabolo is the elderly. Accordingly, we mainly selected participants from the elderly group who regularly performed diabolo exercises at the experimental site. After inquiry and screening, we found all participants to be in good health and without visual or hearing impairments; from this group, we randomly selected 75 participants. All elderly participants who took part in the experiment used wooden diabolos with sound. They had more than 6 months of experience with the diabolo game and maintained regular exercise habits. There was strong heterogeneity in terms of gender (46.7% men and 53.3% women), age (50.7% were 60–70 years old, 33.3% were 70–80 years old, and 16% were older than 80 years), and the number of years of experience with diabolo exercises (21.3% had 1 year or less, 20% had 1 to 2 years, 13.3% had 2 to 3 years, and 45.4% had more than 3 years of experience). The heterogeneity effectively reflects the overall status of the elderly group using diabolos.

#### 2.3.3. Silent State Setting

For this experiment, we needed to set two control states of diabolo exercises, with sound and without sound. The sound principle of a diabolo is related to the whistle in the roulette. The wooden diabolo used by the elderly being tested had a whistle and could make sounds, so it was selected as the experimental equipment for the sound state.

There are two ways to set the silent state. The first choice was to use a plastic diabolo without a whistle. This kind of diabolo will not make sounds. The second was to make a slight modification to the wooden diabolo, sealing the whistle with tape so that there would be no air circulation inside and the diabolo would produce no sound. After investigation, it was decided that the experimental groups would use sound-making wooden diabolos. Silent plastic diabolos and sound-making wooden diabolos are different in their qualities, models, and materials. In order to control the changes in movement caused by these differences, the second method of making the diabolo silent was selected in this experiment.

#### 2.3.4. Control of Environmental Background Noise

Before the experiment, 75 subjects were interviewed with environmental background sounds. The questions were as follows: What do you think is the dominant sound around during diabolo exercise?Do you think the noise of conversation, traffic, etc., around you will interfere with the sound of the diabolo?

According to the data, 72 subjects (96%) believed that the diabolo was the dominant sound when engaging in diabolo movement, and 66 subjects (88%) believed that the environmental background noise would not interfere with the diabolo sound. Interviews with older adults at the venue revealed that when performing diabolo exercises, the diabolo sound is dominant sound they perceive. Environmental background noises (e.g., talking, traffic, birds singing, etc.) have little effect on the sound of the diabolo and are not the primary reason for the difference in the flow state of the elderly when using the diabolo with and without sound. Therefore, we treated environmental background noise as a constant and weak background factor.

#### 2.3.5. Procedure

We conducted the experiment for 2 days on 1 and 2 July 2021. We collected basic information on the characteristics of diabolo soundscape perception and measured the flow state at the experimental site. The sky was clear. According to the survey, diabolo exercises cannot be carried out on rainy days because the diabolo rope is easily affected by moisture. The time when senior citizens engaged in diabolo exercises was mainly sunny mornings. Thus, we carried out the experiment from 6:00 to 10:00 a.m. on sunny days.

We established the flow state in a controlled experiment. The same participants took part in both stages of the experiment, and meditated for 3 minutes before performing the diabolo exercises in each stage to attain the same psychological baseline state. The frequency of diabolo-shaking by all participants was 3 or more times a week, and the measured frequency of the experiment was consistent with the normal frequency of diabolo exercises performed by senior citizens.

The first stage involved evaluating the flow state with the diabolo in the sound state. We conducted the experiment from 6:00 to 10:00 a.m. on 1 July 2021. After the participants finished all of the diabolo exercises, we distributed 75 SFSS questionnaires on the spot and collected 68 valid ones, for a recovery rate of 90.7%. To facilitate the follow-up investigation in the second stage, we marked the participants’ responses in the first stage and coded them. The second stage entailed assessing the flow state with the diabolo in the silent state. We performed the experiment from 6:00 to 10:00 a.m. on 2 July 2021. We followed up with the 75 participants in the first stage; they sealed their diabolo whistle with tape before starting the exercises to prevent it from making sound. After the participants finished all the diabolo exercises, we distributed 75 SFSS questionnaires on the spot and collected 62 valid ones, for a recovery rate of 82.7%. At the end of the experiment, we selected 124 SFSS questionnaires from the 62 participants who had completed it both times as valid for data analysis. Figure 6 shows the specific experimental procedure.

#### 2.3.6. Data Analysis

We used SPSS software (version 20.0, IBM, New York, NY, USA) to process the data. First, we performed a reliability test for the SFSS in both the sound and silent states. Because each dimension of the SFSS has only one item, we carried out a reliability test of the scale as a whole. The Cronbach’s α coefficient for the SFSS was 0.747 (>0.7) in the sound state and 0.738 (>0.7) in the silent state. Reliability was high, thus we could carry out the next round of data analysis. Paired *t*-test was used to analyze the differences in the flow state of participants under the two conditions of the diabolo, with and without sound. Scores of various flow state dimensions and total flow scores in the silent state were significantly lower compared to the sound state (*p* < 0.05 and *p* < 0.001)

## 3. Results

According to the paired *t*-test (Table 2), under the two states of the diabolo (with and without sound), the elderly participants showed significant differences in the eight dimensions of challenge–skill balance, action–awareness merging, clear goals, unambiguous feedback, concentration on the task at hand, sense of control, transformation of time, and autotelic experience, as well as the total score of the flow state (*p* < 0.05 and *p* < 0.001). Only the loss of self-consciousness dimension did not show significant differences. The score of the flow state when using the diabolo in the silent state (25.02) is significantly lower compared to the sound state (35.11), indicating that the flow state of the elderly group will be reduced in silent state.

The value of Cohen’s *d* in Table 2 represents the size of the effect quantity. The larger the value, the greater the difference. According to Cohen’s *d*, the values of the three dimensions of unambiguous feedback, sense of control, and autotelic experience are larger than 0.8 (the critical points for distinguishing small, medium, and large effects are 0.20, 0.50, and 0.80 respectively). Therefore, these three dimensions are the main reasons for the differences in flow state.

## 4. Discussion

### 4.1. Reasons for the Difference in Flow State between Sound and Silent States of Diabolo

As seen by the results in Table 2, the scores of various flow state dimensions and the total flow scores for diabolo exercises in the silent state were significantly lower compared to the sound state (*p* < 0.05 and *p* < 0.001). This result proves that the diabolo soundscape plays an important role in promoting the flow state. The value of mean difference for unambiguous feedback, sense of control, and autotelic experience is relatively large, which was the main reason for the difference in flow state. The direct and indirect influence model of diabolo soundscape perception proposed in Section 2.1 was used to analyze the causes of the differences.

#### 4.1.1. Unambiguous Feedback

Unambiguous feedback means that exercisers are very clear about their performance. The data in Table 2 show that there is a significant difference in the score of this dimension between the silent and sound states (*p* < 0.001), and the score of silent state is decreased by 2.71.

The reason for the difference in this dimension is mainly seen in the indirect perception of the diabolo soundscape. It can be seen from Figure 3 that the sound of the diabolo is an important indicator for measuring the success of playing and to guide the series of actions. When using the diabolo without sound, a lot of feedback and judgment are lost (e.g., whether the diabolo is accelerating in place and running stably), so it is impossible to determine whether one should carry out the next operation.

#### 4.1.2. Sense of Control

The sense of control dimension refers to how fully in control users feel with what they are doing. The data in Table 2 show a significant difference (*p* < 0.001) in the score of this dimension between the silent and sound states, and the score of the silent state decreased by 1.45.

The reason for the difference in this dimension is mainly seen in the indirect perception of the diabolo soundscape. It can be seen from Figure 3 that the sound of the diabolo can coordinate the user’s movements. When the sound was gone, the elderly participants could not coordinate their movements according to the sound; they felt the loss of an important reference basis and their overall rhythm was disturbed. The degree of control over the motion state, rotation speed, and body movement with the diabolo is obviously reduced, resulting in a low flow state.

#### 4.1.3. Autotelic Experience

An autotelic experience refers to enjoying the process of exercise very much. Based on the data in Table 2, the scores of this dimension show a significant difference (*p* < 0.001) between the silent and sound states of diabolo, with the score in the former decreased by 3.53.

The reason for the difference in this dimension is seen in the direct and indirect perceptions of the diabolo soundscape. First, the sound of the diabolo itself has a certain aesthetic value, which can stimulate the auditory nerve of athletes and bring enjoyment. This is a direct effect on the perception of the exercisers. Without the sound, users lost a pleasing soundscape, and their enjoyment was significantly reduced.

Second, regarding indirect perception, the diabolo soundscape is an indicator of success. Hearing the sound represents an excellent operation level, which can enhance the athlete’s confidence and the realization of self-value. Without the sound, the user will not experience enjoyment. The survey of 20 groups of subjects shows that the average number of tourists watching was 14.10 ± 6.25 when diabolo was in the sound state, but there was no crowd in the silent state. This further shows that the sound of diabolo can attract visitors, increase the elderly athletes’ desire to express themselves and have social opportunities, so that they can enhance their self-efficacy and self-esteem through self-expression. However, without the sound, the entertainment and social attributes of diabolo will be reduced, which is not conducive to the flow state.

### 4.2. Mechanism of the Effect of Diabolo Soundscape Perception on Flow State

According to the analysis in Section 4.1, the two-channel influence model of diabolo soundscape perception proposed in Section 2.1 can effectively analyze the reasons for the differences in flow state for the elderly. Therefore, we summarized the influence mechanism of diabolo soundscape perception with regard to the flow state. On the one hand, the diabolo soundscape can directly stimulate people’s aesthetic sense. According to the survey data, 47 (75.8%) of the 62 elderly subjects said they liked the timbre of the diabolo sound very much. Such a sound can make them feel happy. This contributes to the dimension of autotelic experience and affects the flow state of older adults. On the other hand, the diabolo soundscape can help to coordinate movement, indicate steps, symbolize success, and promote social interaction, thus indirectly affecting the three dimensions of the flow state. In this mechanism, the direct and indirect soundscape perception of older adults interact to jointly promote the production of a flow state.

### 4.3. Limitations

The SFSS is mainly administered to professional athletes, and elderly people primarily play diabolo for physical exercise and pleasure. The elderly may not fully understand the questions of the scale, which can lead to some errors in the measurement of flow state.

Just as the role of music in fostering people’s physical and mental health is obvious to both performers and audiences [35], the whistle sound generated during diabolo movements also produces a healing soundscape. This research paves the way for focused study of the diabolo soundscape, which is an active hybrid and healing soundscape, about which there is a lack of evaluation and research. Diabolo movements require exercise equipment in addition to skills and experience. Relative soundscape examples are not as easy to capture as other forms, which is also a factor that restricts investigation and observation. We attempted to examine this topic with special meaning in terms of acoustic landscapes, and to establish a basic framework for diabolo research. Follow-up work will expand the breadth of the sample and the depth of research by tracking and understanding more diabolo fan groups and communities. 

## 5. Conclusions

This study focuses on soundscape perception and proposes a two-channel (direct and indirect) influence model of the effect of the diabolo soundscape on the psychological perception of older players. The Short Flow State Scale (SFSS) was used to measure and compare the flow state of elderly participants when using a diabolo with and without sound, and the two-channel influence model was used to explain the reasons for the differences in flow state. The main conclusions are as follows:(1)To analyze the influence path of diabolo movement in terms of older adults’ psychological perception, a two-channel (direct and indirect) influence model of the effect of the diabolo soundscape on older adults’ psychological perception was constructed.(2)In the silent condition of diabolo movement, the multiple dimensions and total score of flow state were significantly lower than those in the sound condition (*p* < 0.05 and *p* < 0.001), and the three dimensions of unambiguous feedback, sense of control, and autotelic experience had the most important difference. The two-channel influence model can effectively explain the reasons for the differences in older adults’ flow state.

In summary, the diabolo soundscape plays an important role in promoting the flow state of elderly groups. It can affect people’s psychological perception directly and indirectly, which provides a research perspective for studying the role of the soundscape in sports on people’s mental health. At the same time, when building stadiums, architects should pay attention not only to the physical parameters of the soundscape and create a comfortable sound scene environment, but also to the relationship between the soundscape and the site as well as people’s behavior and psychology, so as to create a meaningful soundscape for the crowd.

## Figures and Tables

**Figure 1 ijerph-19-08034-f001:**
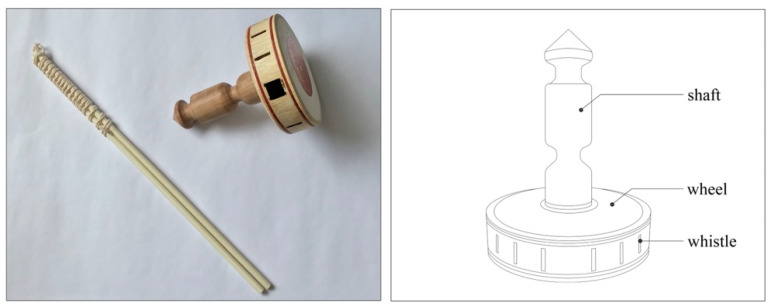
Photo and axonometric drawing of diabolo.

**Figure 2 ijerph-19-08034-f002:**
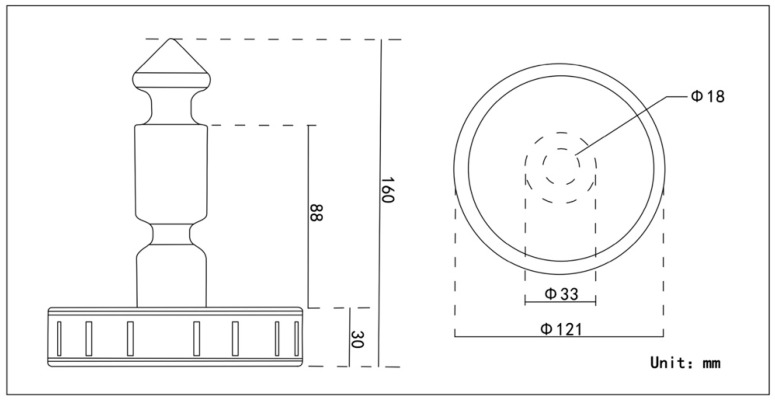
Three-view size chart of diabolo.

**Figure 3 ijerph-19-08034-f003:**
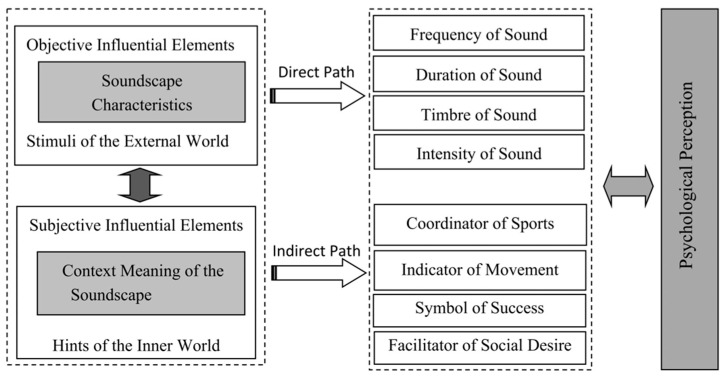
Two-channel (direct and indirect) influence model.

**Figure 4 ijerph-19-08034-f004:**
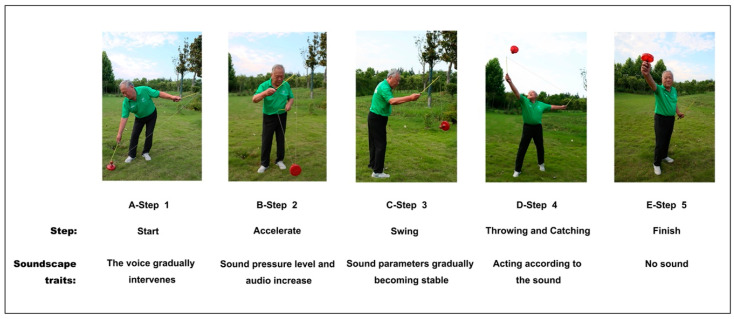
Five basic steps of diabolo exercise.

**Figure 5 ijerph-19-08034-f005:**
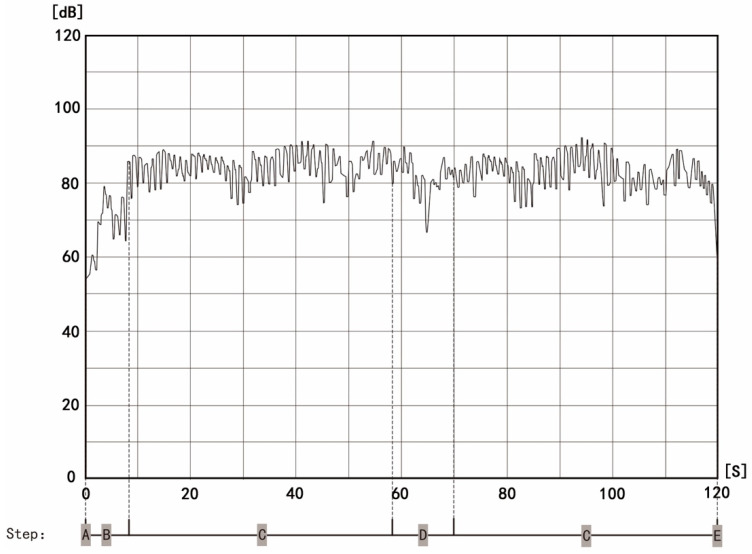
Typical sound levels of diabolo at different steps.

**Figure 6 ijerph-19-08034-f006:**
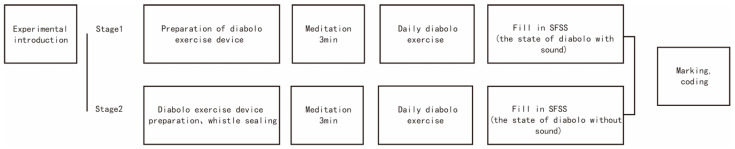
Experimental procedure.

**Table 1 ijerph-19-08034-t001:** Scale dimensions and contents.

Flow State Dimension	Definition of Dimension	Contents
1. Challenge–Skill Balance	In flow, people perceive a balance between the challenges of a situation and their skills, with both operating at a personally high level.	Playing diabolo is difficult, but I can do it very well.
2. Action–Awareness Merging	People are able to attain their intended state as much as possible.	I did all kinds of diabolo techniques without thinking.
3. Clear Goals	Activity goals are well-defined (either set in advance or developed from being involved in the activity), giving people in flow a strong sense of what they are going to do.	I knew exactly what I was going to do.
4. Unambiguous Feedback	In flow state, immediate, clear feedback is received, usually from the activity itself, allowing people to know they are succeeding in the planned goal.	I knew how well I just played.
5. Concentration on Task at Hand	People’s focus is complete and clear when experiencing flow state, without any extraneous thoughts distracting them from the task at hand.	When I was playing diabolo, I was completely focused on that.
6. Sense of Control	People experience a sense of exerting control without actively trying.	I felt completely in control of all the technical moves I was doing.
7. Loss of Self-Consciousness	In flow state, people are completely involved in the activity they are engaged in, becoming part of the action and not caring about success or the values and judgments of others.	I did not care about success or the values and judgments of others.
8. Transformation of Time	In flow state, time is perceptibly altered, sometimes fast and sometimes slow.	I felt like time kept altering, sometimes fast and sometimes slow.
9. Autotelic Experience	An autotelic experience is an intrinsically rewarding experience.	I really enjoyed the experience of playing diabolo.

**Table 2 ijerph-19-08034-t002:** Differences in flow state between sound and silent states of diabolo.

Paired Number	Flow State Dimension	Mean	Standard Deviation	Mean Difference	Cohen’s *d*	*t*	*p*
1	Challenge–skill balance			0.37	0.674	5.309	<0.001
	Sound	3.63	0.81				
	Silent	3.26	0.85				
2	Action–awareness merging			0.10	0.277	2.185	<0.05
	Sound	3.13	1.31				
	Silent	3.03	1.20				
3	Clear goals			0.58	0.725	5.711	<0.001
	Sound	3.89	1.07				
	Silent	3.31	1.29				
4	Unambiguous feedback			2.71	4.252	33.479	<0.001
	Sound	4.61	0.49				
	Silent	1.90	0.30				
5	Concentration on task at hand			0.84	0.770	6.062	<0.001
	Sound	4.79	0.41				
	Silent	3.95	1.00				
6	Sense of control			1.45	1.763	13.882	<0.001
	Sound	3.90	0.84				
	Silent	2.45	0.62				
7	Loss of self-consciousness			0.02	0.127	1.000	>0.05
	Sound	3.31	1.21				
	Silent	3.29	1.19				
8	Transformation of time			0.50	0.745	5.864	<0.001
	Sound	2.95	1.52				
	Silent	2.45	1.31				
9	Autotelic experience			3.53	7.022	55.291	<0.001
	Sound	4.90	0.30				
	Silent	1.37	0.49				
10	Total flow state score			10.10	4.426	34.852	<0.001
	Sound	35.11	5.03				
	Silent	25.02	5.02				

## Data Availability

Not applicable.

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
