# Peer review of "Effects of Soundscape on Flow State during Diabolo Exercise"

_ijerph, 2022, doi:10.3390/ijerph19138034_

Round 1

Reviewer 1 Report

Thanks for the opportunity to review the manuscript titled, " Effects of soundscape perception of sports on flow state:A  case of diabolo exercise". The study investigates how does the sound produce from diabolo affects the flow state, which refers to the sense of enjoyment resulting from engaging in an activity, of the exercisers. The study revealed that the flow state score was significantly higher when sound was produced by diabolo. The authors also proposed a theoretical model to explain the effect of soundscape on the flow state. The study explores the role of sound in exercise psychology, which is a neglected branch of exercise sciences. The authors reviewed the relevant studies, and the study methodology was described in detail. Please find my comments/suggestions below.

Major concerns

1.      The authors stated that written consent was obtained from the participant. But whether the study protocol was approved by an ethics committee remained unknown.

2.       The gentleman shown in figure 3 can be identified. Please provide his consent to the editorial office or mask the face of the gentleman

3.       It remained unknown whether the ‘mechanism’ proposed in section 4.2 is deduced from a statistical model or not. Actually, there is no data/analysis to support the proposed‘ direct effect’ of soundscape on flow state in this study. In conclusion, the authors state that ‘Based on the analysis of direct perception and indirect perception ……. perception on flow state was summarized.’ In my opinion,  the analyses conducted in this manuscript cannot support the validity of the proposed model. The direct and indirect effects should be assessed statistically using a statistical tool such as structural equation modelling.

Other comments

4.       I would suggest revising the title to ‘Effects of soundscape perception on flow state diabolo exercise’.

5.       The authors selected three items of the SFSS to explain the effect of sound on flow state because these items accounted for 76% of the differences in the SFSS scale. The cut-off threshold is arbitrary. Perhaps the authors can use a scree  plot to justify their decision

6.       Are there any pre-defined inclusion and exclusion criterion ?

7.       Line 374 The description regarding the experimental error is too vague. Please suggest the potential bias of using the questionnaire in this population, instead of stating ‘hence is, there are some error’. Besides, this section should be combined with section 4.4 Limitation.

8.       Line 385 The future research direction is non-specific. Please suggest future research that is derived from the results of the current study.

9.       Please include effect size in table 2. Besides, don’t use 0.000 in the column of P value, use <0.001 instead.

Author Response

Dear Editors and Reviewers:

Thank you for your letter and for the reviewers' comments concerning our manuscript entitled " Effects of soundscape perception of sports on flow state:A case of diabolo exercise" (ID: ijerph-1729685). Those comments are all valuable and very helpful for revising and improving our paper, as well as the important guiding significance to our researches. We have studied comments carefully and have made correction which we hope meet with approval. Revisions to the manuscript are marked up using the “Track Changes” function in the paper. The main corrections in the paper and the responds to the reviewer's comments are as flowing:

Responds to  Reviewer 1 Comments:

  1. Response to comment 1:Statement of the Approval of Ethical Review

Response:

Thanks for the reviewer's comment. At present, in the disciplines of urban and rural planning, landscape architecture and other architectural disciplines, the ethical reviews of social investigation and research for the public and residents are generally conducted by the professor committees of colleges and universities because they do not involve the complex, sensitive or related safety content in the medical field. Hence, we submitted the statement of the ethical review approved by Professor Committee of School of Architecture, Harbin Institute of Technology to the editorial office.

  1. Response to comment 2:Written consent of the subject in Figure 3

Response:

Considering the reviewer's suggestion, we provided the consent of the subject in figure 3 to the editorial office.

  1. Response to comment 3:The analyses conducted in this manuscript cannot support the validity of the proposed model in section 4.2.

Response:

Thanks for the reviewer's comment. Considering that there is no data/analysis to support the proposed ‘direct effect’ of soundscape on flow state in this study, we deleted this model and revised chapter 4.2 of the article to strengthen the logic of the argument. (Line 398)

  1. Response to comment 4:Suggestion on revision of manuscript title

Response:

Line 2, we have revised the title to ‘Effects of soundscape perception on flow state diabolo exercise’.

  1. Response to comment 5 and 6: The authors selected three items of the SFSS to explain the effect of sound on flow state because these items accounted for 76% of the differences in the SFSS scale. The cut-off threshold is arbitrary. Perhaps the authors can use a scree plot to justify their decision. Are there any pre-defined inclusion and exclusion criterion?

Response:

Inspired by the reviewer's comment 9, we have chosen a more reasonable way to compare differences(Please see lines 324-329 for details). We added Cohen's d value in Table 2 to measure the effect size. By comparing the value of Cohen's d, we can compare the difference size of the dimensions of SFSS. The Cohen's d values of the three dimensions ("unambiguous feedback", "sense of control" and " autotelic experience”) are larger than 0.8 (the critical points for distinguishing small, medium and large effects are 0.20, 0.50 and 0.80 respectively). Therefore, these three dimensions are the main reasons for the differences in flow state.

  1. Response to comment 7: Line 374 The description regarding the experimental error is too vague. Please suggest the potential bias of using the questionnaire in this population, instead of stating ‘hence is, there are some error’. Besides, this section should be combined with section 4.4 Limitation.

Response:

Considering the reviewer's suggestion, we gave a specific explanation for the experimental error and combined section 4.3 with section 4.4 Limitation (Line 399).

  1. Response to comment 8: Line 385 The future research direction is non-specific. Please suggest future research that is derived from the results of the current study.

Response:

Thanks for the reviewer's comment. In the conclusion part, we added the future research that is derived from the results of the current study. (Please see lines 436-443 for details)

  1. Response to comment 9: Please include effect size in table 2. Besides, don’t use 0.000 in the column of P value, use <0.001 instead.

Response: We added Cohen's d value in Table 2 to measure the effect size. We also used <0.001 instead of 0.000, <0.05 instead of 0.033 and >0.05 instead of 0.321 in the column of P value (Lines 324-329).

Reviewer 2 Report

Review comment

This manuscript entitled “Effects of soundscape perception of sports on flow state: A case of diabolo exercise” primarily aimed to explore relationships between the soundscapes of sport itself and the mental health of exercisers and to provide a reference for how to improve the mental health of exercisers by improving the soundscape. The authors bring an interesting study, but there are still some problems that cannot up this study to a publishing level. Some suggestions are listed in the specific comments below.

Specific comments:

1.     Abstract, the description of the method section is unclear, and the results section does not provide sufficient information about the experimental results of this study. Please avoid using subjective words, such as ‘creatively’.

2.     ‘The results show that the perception of diabolical soundscape can directly and indirectly affect athletes' flow state.’, please recheck ‘diabolical’. The word ‘athletes’ may be misleading; it would be better to use ‘older adults’ here.

3.     Keywords, please modify and improve the quality of the keywords as this will assist others when they are searching for information on your research topic. Avoid using ‘diabolo exercise’, ‘flow state’, and ‘soundscape’ since they appear in the title.

4.     Introduction, ‘…but the soundscape of sport itself (such as the soundscape of diabolo exercise in China) is also valuable for research.’, ‘To the best of the authors knowledge, no review has been performed so far on the association ...’, the lack of relevant studies clearly does not justify the conduct of this study. The necessity and significance of this study need to be further clarified.

5.     ‘…to innovate the two-channel influence model (direct influence and indirect influence) of diabolo soundscape perception’, What do ‘direct influence’ and ‘indirect influence’ mean respectively?

6.     Materials and Methods, ‘Diabolo and its soundscape’, ‘Diabolo is a traditional Chinese sport and a state-level, non-material cultural heritage…’, the introduction of Diabolo in this section is too verbose and unnecessary (or would be more appropriate in the introduction section), please simplify it further.

7.     ‘Evaluating flow state’, ‘applied the Chinese version of Liu Weina’s SFSS’, ‘The SFSS has high reliability, and many scholars use it to study people s flow state when they engage in physical exercise.’, Please add references where appropriate.

8.     ‘Data analysis’, please provide more details about the statistical analysis in this study, such as the significance level.

9.     Results, ‘This indicates that the flow state of the elderly group during the silent state of diabolo exercises was impaired, which is not conducive to promoting the mental health of older adults.’, please present objective experimental results in this section, rather than subjective analysis.

10.  Discussion, ‘At the same time, the sound of diabolo can also attract visitors, increase the desire of expression and social opportunities of the elderly athletes, so that they can enhance their self-efficacy and self-esteem through self-expression.’, in this study, did the authors conduct a survey on the number of visitors attracted by the performance of diabolo?

11.  Conclusion, please simplify the conclusion further.

12.  English is understandable but not suitable for an international scientific journal; please have the language reviewed by a native speaker.

13.  In summary, please ensure that your manuscript is prepared correctly (without any grammatical and spelling mistakes) and formatted before submitting a revision.

Author Response

Dear Editors and Reviewers:

Thank you for your letter and for the reviewers' comments concerning our manuscript entitled " Effects of soundscape perception of sports on flow state:A case of diabolo exercise" (ID: ijerph-1729685). Those comments are all valuable and very helpful for revising and improving our paper, as well as the important guiding significance to our researches. We have studied comments carefully and have made correction which we hope meet with approval. Revisions to the manuscript are marked up using the “Track Changes” function in the paper. The main corrections in the paper and the responds to the reviewer's comments are as flowing:

Responds to Reviewer 2 Comments:

  1. Response to comment 1: Abstract, the description of the method section is unclear, and the results section does not provide sufficient information about the experimental results of this study. Please avoid using subjective words, such as ‘creatively’.

Response: Thanks for the reviewer's comment. We revised the abstract, strengthened the description of the method section, and added relevant data of experimental results, making the abstract more logical on the whole. At the same time, we deleted some words with strong subjectivity, such as ‘creatively’. (Please see lines 22-34 for details).

  1. Response to comment 2: Please recheck ‘diabolical’. The word ‘athletes’ may be misleading; it would be better to use ‘older adults’ here.

Response: We are very sorry for our incorrect writing. We changed ‘diabolical’ to ‘diabolo’ and ‘athletes’ to ‘older adults’.

  1. Response to comment 3: Please modify and improve the quality of the keywords.

Response: Considering the reviewer's suggestion, we have modified the keywords. The modified keywords are ‘diabolo’; ‘exercise psychology’; ‘flow state’; ‘soundscape’; ‘mental health’ and ‘older adults’ (line 37).

  1. Response to comment 4: Introduction, the necessity and significance of this study need to be further clarified.

Response: Thanks for the reviewer's comment. We have revised this paragraph from the perspective of the concept of soundscape to highlight the significance and importance of this study. (Please see lines 80-90 for details).

  1. Response to comment 5: What do ‘direct influence’ and ‘indirect influence’ mean respectively?

Response: We quoted some references to explain the direct and indirect influence of diabolo soundscape to people's psychological perception (Line 165-177).

  1. Response to comment 6: The introduction of Diabolo in this section is too verbose and unnecessary.

Response: Considering the reviewer's suggestion, we have simplified the content and added some contents to the introduction section (Line 91).

  1. Response to comment 7: The SFSS has high reliability, and many scholars use it to study people s flow state when they engage in physical exercise. Please add references where appropriate.

Response: Thanks for the reviewer's comment. We have added relevant references (Line 212).

  1. Response to comment 8: Please provide more details about the statistical analysis in this study, such as the significance level.

Response: Thanks for the reviewer's comment. We have provided more details about the statistical analysis in this study (Line 311-313).

  1. Response to comment 9: ‘This indicates that the flow state of the elderly group during the silent state of diabolo exercises was impaired, which is not conducive to promoting the mental health of older adults.’, please present objective experimental results in this section, rather than subjective analysis.

Response: Considering the reviewer's suggestion, we modified this sentence according to the results of the experiment (Line 322).

  1. Response to comment 10: In this study, did the authors conduct a survey on the number of visitors attracted by the performance of diabolo?

Response: We added data from a survey on the number of visitors attracted by the performance of diabolo in order to better prove our point (Line 378-380).

  1. Response to comment 11: Please simplify the conclusion further.

Response: Considering the reviewer's suggestion, we simplified the conclusion (Line 414).

  1. Response to comment 12-13: Please have the language reviewed by a native speaker and ensure that your manuscript is prepared correctly.

Response: Thanks for the reviewer's comments. We have sent the manuscript to native speakers of English for revision. In addition, the manuscript was submitted to AJE company for editing. (The modification period is about one week, revised manuscript will be uploaded later.)

Reviewer 3 Report

This study does not have a very strong rationale and seems to have very little application. The writing needs improvement and further revisions in this area are required. There is a lot of repeating of information throughout the manuscript. The writing also needs to be more concise. Below are further comments.

Lines 16-17: A very brief definition of ‘soundscape’ would be helpful to the reader.

Lines 20-22: What is the aim of the study?

Lines 28-29: Still unsure about the rationale for the study? I suggest mentioning why this study is important.

Line 38: Why does soundscape research need to focus on these topics?

Line 75: What is ‘diabolo’? Please define.

Line 113: What sport?

Line 116: ‘..measured it with..’

Line 118: This sentence is incomplete. Please revise.

 Line 136: Please do not use contractions in you manuscript e.g. doesn’t should be does not.

Line 147: ‘..the diabolo’s…”

2.1 – This information is highly inappropriate for this section. You should be speaking about the participants, the study design and intervention, what measurements were performed, how the measures were performed etc.

Figure 3: ‘acting according to the sound’

Figure 4: ‘Typical sound-level of the Diabolo at different-steps’

Line 171: ‘Diabolo movement is a whole composed…’ – English grammar needs improvement here.

Line 174: ‘…scape is different in different stages…’ – Please revise the word choices.

2.2 – Again this information is inappropriate for this section.

Figure 5: ‘Regulate Endocrine System’ and ‘Facilitator of Social…’of what?

Figure 6. ‘SFSS’ please delete ‘scale’

Line 282:SFSS’ please delete ‘scale’ plus in other places throughout the manuscript. It should be just ‘SFSS’.

Figure 7: What is a ‘facilitator of social’? – this does not make sense?

Line 409: ‘In summary,…”

Author Response

Dear Editors and Reviewers:

Thank you for your letter and for the reviewers' comments concerning our manuscript entitled " Effects of soundscape perception of sports on flow state:A case of diabolo exercise" (ID: ijerph-1729685). Those comments are all valuable and very helpful for revising and improving our paper, as well as the important guiding significance to our researches. We have studied comments carefully and have made correction which we hope meet with approval. Revisions to the manuscript are marked up using the “Track Changes” function in the paper. The main corrections in the paper and the responds to the reviewer's comments are as flowing:

Responds to Reviewer 3 Comments:

  1. Response to comment:A very brief definition of ‘soundscape’ would be helpful to the reader. (Lines 16-17)

Response:

Considering the Reviewer's suggestion, we have added a brief definition of ‘soundscape’. ‘Soundscape’ is defined as an ‘acoustic environment as perceived or experienced and/or understood by a person or people, in context’ (lines 14-15).

  1. Response to comment:What is the aim of the study ? (Lines 20-22)

Response:

The purpose of this study is to explore the influence mechanism of soundscape on the mental health of exercisers/athletes in sports, so as to better guide the shaping of soundscape in sports venues (lines 22-24).

  1. Response to comment:Still unsure about the rationale for the study? I suggest mentioning why this study is important.

Response:

Thanks for the reviewer's comments. We have revised this paragraph from the perspective of the concept of soundscape to highlight the significance and importance of this study. (Please see lines 80-90 for details).

  1. Response to comment:Why does soundscape research need to focus on these topics?

Response:

Thank you very much for your question. According to our research on relevant literature, only focusing on the soundscape physical parameters can not really promote people to integrate into this scene. Therefore, it is very important to pay attention to how soundscape connects people and environment, and there is a lack of research in this field.

  1. Response to comment:What is ‘diabolo’? Please define.

Response:

Diabolo is a traditional Chinese sport with a multidimensional soundscape. We have revised this part and put the introduction of diabolo introduction section (Line 91).

  1. Response to comment:Line 113: What sport?

Response:

I am very sorry that this sentence is not clearly stated. We have modified this sentence (Line 151).

  1. Response to comment:Line 116: ‘..measured it with..’

Response:

We used a scale to measure flow state. I am very sorry that this sentence is not clearly stated. We have modified this sentence (Line 153).

  1. Response to comment:Line 118: This sentence is incomplete. Please revise.

Response:

Thanks for the reviewer's comments. We have modified this sentence (Line 155).

  1. Response to comment:Line 136: Please do not use contractions in you manuscript e.g. doesn’t should be does not.

Response:

We revised the contractions in the paper,and replaced doesn’t with does not(Line 101-102), replaced it’s with it is (Line 101).

  1. Response to comment:Line 147: ‘..…”

Response:

We revised the contractions in the paper, and replaced the diabolo’s with the sound of diabolo (Line 182).

  1. Response to comment:2.1 – This information is highly inappropriate for this section. You should be speaking about the participants, the study design and intervention, what measurements were performed, how the measures were performed etc.

Response:

Thanks for the reviewer's comments. We have modified chapter 2.1 and adjusted the content of chapter 2.1 to the introduction section (Line 91).

  1. Response to comment:Figure 3: ‘acting according to the sound’

Response:

Considering the reviewer's suggestion, we have changed the name of Figure 3 to ‘acting according to the sound’. (Line 206)

  1. Response to comment:Figure 4: ‘Typical sound-level of the Diabolo at different-steps’

Response:

Considering the reviewer's suggestion, we have changed the name of Figure 4 to ‘Typical sound-level of the Diabolo at different-steps’. (Line 210)

  1. Response to comment:Line 171: ‘Diabolo movement is a whole composed…’ – English grammar needs improvement here.

Response:

Thanks for the reviewer's comments. We have modified this sentence:

Diabolo movement is composed of a series of movements as a whole, and sound is the important content connecting each link. (Line 188)

  1. Response to comment:Line 174: ‘…scape is different in different stages…’ – Please revise the word choices.

Response:

Thanks for the reviewer's comments. We have modified the word:

Diabolo soundscape is different in different stages of diabolo movement. (Line 191)

  1. Response to comment:2.2 – Again this information is inappropriate for this section.

Response:

Thanks for the reviewer's comments, we have modified this paragraph and focused on deriving the direct-and-indirect influence model of diabolo soundscape perception. This is part of our research design. The revised paragraph is suitable for ‘materials and methods’. (Line 164)

  1. Response to comment:Figure 5: ‘Regulate Endocrine System’ and ‘Facilitator of Social…’of what?

Response:

Thanks for the reviewer's comments. We modified Figure 5 and deleted the impact of diabolo on people's physiology (the physiological impact has nothing to do with this study), forming a new figure 3. (Line 207)

  1. Response to comment:Figure 6. ‘SFSS’ please delete ‘scale’. Line 282: ‘SFSS’ please delete ‘scale’ plus in other places throughout the manuscript. It should be just ‘SFSS’.

Response:

We are very sorry for our negligence of abbreviation of scale name. We have checked the abbreviation of this scale in the full text and deleted the word "scale" in the original text. (Line 311;314;316;317)

  1. Response to comment:Figure 7: What is a ‘facilitator of social’? – this does not make sense.

Response:

Considering that there is no data/analysis to support the proposed ‘direct effect’ of soundscape on flow state in this study, we deleted this model (Figure 7) and revised chapter 4.2 of the article to strengthen the logic of the argument. (Line 398)

  1. Response to comment:Line 409: ‘In summary,…”

Response:

Thanks for the reviewer's comment. We have replaced 'In sum' with 'In summary’. (Line 449)

Round 2

Reviewer 1 Report

In general, the content of the revised manuscript has been improved. And the authors addressed my comments. However, I believe that English editing is necessary before publishing.

Reviewer 2 Report

The authors has completed and satisfied all the recommendations that I raised and stated.

Reviewer 3 Report

Well done on addressing my concerns.